# Monodisperse Fluorescent Polystyrene Microspheres for *Staphylococcus aureus* Aerosol Simulation

**DOI:** 10.3390/polym15173614

**Published:** 2023-08-31

**Authors:** Siyu Lu, Fan Li, Bo Liu, Kun Yang, Feng Tian, Zhi Cheng, Sheng Ding, Kexin Hou

**Affiliations:** 1Medical Support Technology Research Department, Systems Engineering Institute, Academy of Military Sciences, People’s Liberation Army, Tianjin 300161, China; luuuuuu1026@163.com (S.L.); lyclb2020@163.com (B.L.); yangkuntianda@163.com (K.Y.); chengzh@npec.org.cn (Z.C.); 18317101681@163.com (S.D.); hkxnefu@163.com (K.H.); 2National Bio-Protection Engineering Center, Tianjin 300161, China

**Keywords:** *Staphylococcus aureus*, polystyrene, microsphere, monodisperse, aerosols, simulation

## Abstract

*Staphylococcus aureus* (SA) is one of the most common causes of hospital-acquired infections and foodborne illnesses and is commonly found in nature in air, dust, and water. The spread and transmission of SA aerosols in the air has the potential to cause epidemic transmission among humans and between humans and animals. To effectively provide the timely warning of SA aerosols in the atmosphere, the identification and detection of SA aerosol concentrations are required. Due to their homogeneous physicochemical properties, highly monodisperse submicron polystyrene (PS) microspheres can be used as one of the simulants of SA aerosols. In this study, 800 nm monodisperse fluorescent PS (f-PS) microspheres with fluorescence spectra and particle size distribution similar to those of SA were prepared. The 800 nm monodisperse f-PS microspheres had a fluorescence characteristic peak at 465 nm; in aerosols, 800 nm monodisperse f-PS microspheres with a similar particle size distribution to that of SA were further verified, mainly in the range of 500 nm–1000 nm; finally, it was found that the f-PS microspheres still possessed similar fluorescence characteristics after 180 days. The f-PS microspheres prepared in this study are very close to SA in terms of particle size and fluorescence properties, providing a new idea for aerosol analogs of SA.

## 1. Introduction

Bioaerosols are aerosols composed of any kind of biological particle [1,2] with an aerodynamic diameter of 100 μm or less [3,4,5,6,7]. Aerosols are responsible for the airborne transport of microorganisms in the air and consist of small particles called droplet nuclei (1–100 μm) or droplets (>100 μm). Droplet nuclei can remain in the air for hours, travel long distances, and contaminate surfaces by falling. Bacteria such as *Bacillus*, *Pseudomonas*, *Staphylococcus aureus* (SA), *Micrococcus*, and *Acinetobacter* have been identified as predominant in bioaerosols [8,9,10,11], and the dominant fungi are *Penicillium*, *Aspergillus*, *Candida*, *Cladosporium*, and *Rhodotorula* [8,12,13,14,15]. The diffusion and spread of biological aerosols in the air have the potential to cause epidemic transmission among humans and between humans and animals, and in some ways are more harmful than non-biological aerosols. Therefore, the detection of bioaerosols for early warning is necessary, and at present, there are a variety of bioaerosol monitors on the market for the detection of bioaerosols for early warning [1,2]. However, in the development, calibration, and performance testing of aerosol detection devices and the real protection performance testing of bio-protective equipment, it is impractical to use real microbial aerosols with infectious properties every time, and bio-aerosol leakage can easily occur. Therefore, the development of simulants characterized by pathogenic microorganisms can make the calibration and testing of aerosol devices easier, faster, and safer.

Monodisperse polystyrene (PS) microspheres, which have stable properties and are not difficult to prepare, have the potential to be bioaerosol mimics. Monodispersity usually refers to the uniform particle size or diameter distribution of the microspheres, and monodisperse PS microspheres have a wide range of applications in several technological fields due to their homogeneous physicochemical properties [16,17,18,19,20,21,22]. PS, as a polymeric material with high light transmittance, is one of the main synthesized materials for fluorescent microspheres, which have been widely used as a reference reagent for proteins, gene vectors, and absolute cell counting [23,24].

There is a wide variety of preparation methods for PS microspheres, and different preparation methods can be selected according to different particle sizes. Regarding the synthesis of submicron PS microspheres in addition to the traditional methods such as dispersion polymerization, microfluidic technology is now also used to prepare monodisperse microspheres. There are two main types of devices for generating emulsions in microfluidic platforms: T-junctions [25,26,27] and flow-focused nozzles [28]. Both methods can generate monodisperse particles and both offer flexibility in the size of the emulsions generated. In addition, monodisperse poly-ε-caprolactone was prepared by Alemrayat et al. using vibrating orifice aerosol generator technology for the first time in 2018 [29]. Electrohydrodynamic atomization, a subfield of fluid mechanics, is also used in the preparation of monodisperse microspheres. The technique applies an electrical charge to a viscous liquid through a capillary nozzle, leading to the formation of droplets and further atomizing particles to possess varying sizes from hundreds of micrometers (μm) to several tens of nanometers (nm) [30].

SA is one of the most common causes of hospital-acquired infections and foodborne diseases. Its toxin and highly toxic protease usually circulate in the host’s blood vessels, leading to life-threatening diseases. Standard identification methods include bacterial culture methods, which may take several days. The current bioaerosol monitor enables the instrument to monitor biological particles in the air in real time [31,32]. However, when calibrating the SA aerosol monitoring instrument, it is necessary not only to identify and count SA, but also to calibrate the particle size. This requires that the prepared standard can not only be excited to produce fluorescence at a certain wavelength, but also be strictly monodisperse in terms of particle size. At present, two types of standards are mainly used for the commissioning and calibration of bioaerosol monitors: bacteria and fluorescent microspheres. Compared with the bacterial standard, the fluorescent microsphere has the characteristics of good controllability, high stability, low toxicity, and simple operation. However, in order to find mimic microspheres that conform to SA, three conditions must be met: high safety, similar particle size distribution, and fluorescence characteristics.

It has been reported that the particle size of SA is about 800–1000 nm [33,34], and at present, there are few studies simulating a single SA with different applications. In practical applications, aerosols are mostly colony-dominated with a wide range of particle sizes, which is not conducive to accurate identification and differentiation. In this study, monodisperse fluorescent polystyrene (f-PS) microspheres with diameters and fluorescence properties similar to SA were obtained by dispersion polymerization and serial screening techniques, and the mimics effectively mimicked SA in terms of particle size, fluorescence eigen signal, and aerosol mimicry. To the best of our knowledge, this is the first monodisperse f-PS microsphere reported for SA aerosol mimicry.

## 2. Materials and Methods

### 2.1. Materials

Samples of 2,2-azobisisobutyronitrile (AIBN, A104256) and polyvinylpyrrolidone (PVP, P274371) were ordered from Aladdin Co., Ltd. (Shanghai, China); styrene (St, S817904) was purchased from Macklin Co., Ltd. (Shanghai, China); anhydrous aluminum chloride (206911) and hydrochloric acid solution (W530574) were ordered from Sigma-Aldrich Co., Ltd. (St. Louis, MO, USA), and 0.65 μm low-binding Durapore PVDF membrane (UFC40DV25) was purchased from Merck Co., Ltd. (Rahway, NJ, USA). Acetone (48358) was purchased from Supelco Co., Ltd. (Bellefonte, PA, USA).

### 2.2. Methods

#### 2.2.1. Synthesis of Functional Submicron-Size Microspheres

x wt% St (x = 6, 10, 14), y wt% PVP (y = 1, 3, 5), z wt% AIBN (z = 0.6, 1, 1.4) were added to each 100 mL of ethanol solution, the volume ratio of water to ethanol in said ethanol solution was m:36 (m = 0, 4, 9), and the reaction temperature was n °C (*n* = 60, 70, 80). Each 100 mL of the configured ethanol solution was dissolved into the collective thermostatic heating magnetic stirrer (DF-101S), nitrogen was passed to exclude the oxygen in the system, the collective thermostatic heating magnetic stirrer was turned on for polymerization reactions to obtain PS submicron microsphere dispersion, the PS submicron microsphere was centrifuged and collected, suspended with 5 mL of anhydrous ethanol, and 5 μL of aminosilane was added. The solution was stirred for 1 h, washed with an equal volume of anhydrous ethanol, and collected. Chloromethylated fluorescent PS (f-PS) microspheres were prepared by Friede—Crafts alkylation [35]. Then, 10 g of PS microspheres were dissolved in 100 mL of chloromethyl methyl ether for 60 min, and then 1.2 wt% aluminum trichloride was added to the mixture. After stirring at 50 °C for 12 h, the mixture was washed with acetone, hydrochloric acid solution, anhydrous ethanol, and deionized several times, and then collected by centrifugation.

#### 2.2.2. Screening of PS Microspheres

Glycerol/water solutions with certain gradient mass ratios of 10%, 25%, 50%, etc., were configured and their density and viscosity calculated. The prepared PS microsphere samples were added to the glycerol/water solution to determine the glycerol/water mass ratio at which the microspheres could be suspended in the solution for a long time, when the density of the liquid was the same as the density of the microspheres. The solution was then replaced with a suitable glycerol/water mass ratio, the time required for the target microspheres to settle to the bottom was calculated according to the equation, the sample was settled, and after reaching the calculated time, the upper layer of small microspheres was removed; this was repeated several times until the upper layer was close to clarification. The sample was settled again, and at an appropriate time before the calculated time, the upper liquid microspheres were collected and the substrate microspheres removed. This was repeated several times until the substrate had no obvious microsphere precipitation, and finally, the target particle size microspheres were obtained.

The synthesized microspheres always require several separations to obtain polymer microspheres of uniform particle size. The gradient sedimentation method can easily remove a portion of very-large-sized microspheres and agglomerates of microspheres after the reaction by sieving, and another portion of very-small-sized microspheres can be removed during centrifugal washing. However, the microspheres with particle size of 500 nm–50 µm were difficult separate and purify simply by using conventional methods such as centrifugation. Therefore, we further used 0.65 μm low-binding Durapore PVDF membrane (UFC40DV25) for the further sieving of the PS microspheres. The previously obtained target-sized microspheres were added to the 0.65 μm low-binding Durapore PVDF membrane, centrifuged at 5000 rpm for 10 min, and resuspended in the upper filter with ethanol. This was repeated three times to obtain the target-sized microspheres.

#### 2.2.3. Characterization of Functional Submicron-Size Microspheres

The images of different PS microspheres were observed by scanning electron microscopy (Tescan MIRA LMS, Resolution: 0.9 nm @ 15 Kv (secondary electron image); 2.0 nm @ 30 Kv (backscattered electron image), accelerating voltage: 200 v–30 kv, probe beam current: 1 pA–100 nA, stability better than 0.2%/h, magnification: 8×–100,000×), and the average particle size and particle size distribution of the microspheres were analyzed by dynamic light scattering (DLS, Nano ZS ZEN3600). A trace sample was glued directly onto the conductive adhesive and sprayed with gold using a Quorum SC7620 sputter coater for 45 s at 10 mA, followed by a TESCAN MIRA LMS scanning electron microscope to photograph the sample morphology, which was accelerated at 3 kV. In probability theory and statistics, the coefficient of variation (CV), also known as relative standard deviation (RSD), is a standardized measure of the dispersion of a probability distribution or frequency distribution. The CV is equal to the standard deviation (SD) divided by the mean. The particle size distribution was evaluated by CV: the larger the CV, the wider the particle size distribution. For ideal monodisperse microspheres, the CV is 0. For most commercially available monodisperse microspheres, this value usually ranges from 10% to 20%. If the CV is higher than 20%, the particle size distribution is too wide.

#### 2.2.4. Fluorescence Spectrum and Particle Size Distribution of SA

Standard strains of frozen preserved bacteria (ATCC 29213) were revived by scribing on nutrient agar and incubated in a constant temperature incubator (ZXDP-B2160) at 37 °C for 18–24 h. Individual SA colonies were picked from the plates, inoculated into nutrient broth, shaken at a constant temperature of at 37 °C, and incubated for 18–24 h. The bacterial solution was taken and inoculated onto Baird–Parker medium at 37 °C until colonies grew, and then 10 colonies were randomly picked for Gram staining and observed under a microscope. The bacteria were arranged in the form of staphylococci, without budding cells and without pods, with diameters of about 0.5–1.0 µm. Then, 2 mL of the bacterial suspension was placed in the counting cell of the bacterial counting plate and counted by microscopic observation. The number of bacteria contained in the sample was calculated based on the number of bacteria within the counting plate scale. A solution containing 10^8^ SA per mL was selected for subsequent experiments, and the fluorescence spectra of SA were observed using a steady-state/transient fluorescence spectrometer (Edinburgh FLS1000). The average particle size and size distribution of the microspheres were determined by a nanoparticle size and zeta potential analyzer (Nano ZS ZEN3600).

#### 2.2.5. Fluorescence Spectrum and Particle Size Distribution of f-PS Microspheres in Aerosols

The monodisperse PS microspheres and SA were atomized to form an aerosol environment with uniform and stable concentrations. The sampler was placed in a measurement chamber, the aerosol particle concentrations upstream and downstream of the sampler were measured with a portable particle counter (TSI 9110), and the particle size distribution was analyzed.

The evaluation system was divided into three main parts. The first part atomizes monodisperse PS microspheres and SA aerosol particles and regulates the concentration of particles in the chamber by controlling the flow rate of atomization and dilution gases; the second part mixes the particles produced by the first part atomization with air. The planktonic bacteria sampler to be measured was placed in the measurement chamber, which was equipped with a high-efficiency filter at the bottom to realize the gas exchange with the outside world and avoid the interaction between the particles in the chamber and the outside air. The third part, through the portable particle counter and the pump under the measuring chamber, together make the gas flow into the sampler meet its working flow requirements and perform the measurement.

## 3. Results

### 3.1. Analysis of PS Microsphere Preparation and Screening Process

#### 3.1.1. Effect of Different St Concentrations on the Particle Size of Microspheres

The group determined the optimum reaction conditions by adjusting the conditions for dispersion polymerization as follows (Table 1): 10 wt% St (Appendix A), 3 wt% PVP (Appendix A), 1 wt% AIBN (Appendix A) were added to each 100 mL of ethanol solution. The volume ratio of water to ethanol in said ethanol solution was 1:4 (Appendix A) and the reaction temperature was 70 °C (Appendix A).

The particle sizes of the microspheres prepared in this experiment using 6, 10, and 14 wt% styrene were 560, 722, and 779 nm, respectively (Appendix A), and the particle size distribution of the microspheres became larger with increasing styrene dosage, with the best styrene concentration being 14 wt%. The monodisperse PS microspheres were not obtained at too low a styrene concentration.

In the formation of primary nuclei and primary particles, the viscosity of the system at different PVP concentrations was different, and the interfacial energy between the primary nuclei and the continuous phase was also different, so the diameters of the primary particles formed in different systems were also different. As shown in Appendix A, in systems with lower PVP concentrations (1–5 wt%), the system viscosity was lower, the surface tension was lower, and the diameter of the primary nuclei formed was larger; however, the suspension and dispersion ability of the system was weaker and the chances of collision and fusion of the primary nuclei formed were higher. In this experiment, the 3 wt% PVP concentration was chosen as the optimum concentration due to the need for a specific particle size.

The particle size of the aggregates will have a Gaussian distribution due to collisions and the deviation becomes larger as the particle size increases (Appendix A); most studies usually choose an initiator of 1–2 wt%. In this study, an AIBN concentration of 1% was used due to the need for particle size.

Water is a good solvent and the presence of water does not affect the solubility of PVP. However, for styrene monomers, the addition of water reduces the solubility and, thus, the solubility of the reaction medium for the polymer formed, resulting in a reduction in the critical chain length for nucleation, an increase in the number of primary particles, and a reduction in particle size. As shown in Appendix A, the particle size of the PS microspheres gradually increased from 329 nm to 821 nm as the water/alcohol ratio was gradually increased from 0% to 20%. However, the coefficient of variation decreased from ~18% to ~15% and then gradually increased to ~17%. In this study, the water/alcohol concentration ratio was 20% due to the requirement for particle size.

The appropriate temperature has a very important effect on the particle size and dispersion effect of PS microspheres, as the temperature directly affects the decomposition rate of the initiator and the critical chain length of the polymer chains. As shown in Appendix A, at a lower temperature of 60 °C, the system was slightly less soluble for the primary nuclei, the system viscosity was larger, and the fusion and dispersion between the microspheres and the material exchange process was less, so the particle size distribution of the microspheres became wider. And at 70 °C, the state of the system was more stable and the decomposition rate of the initiator was more suitable, so the particle size dispersion of the formed microspheres was more uniform and the number of secondary particles was lower. At a higher temperature of 80 °C, the decomposition process of the initiator was very intense, and a high concentration of local free radicals was formed, which, in turn, led to a high concentration of local primary nuclei, and, thus, a large number of primary particles can be formed (similar to the effect of AIBN on the microspheres). However, the particle size distribution of the microspheres became wider because the dispersion was not uniform enough, and the system viscosity decreased at high temperatures when the dispersant protection ability became poor. Therefore, 70 °C was chosen as the heating temperature for this experiment.

#### 3.1.2. Effect of Different Screening Methods on the Particle Size Distribution of Microspheres

In most polymerization reactions for the preparation of microspheres, there are also other reaction conditions that affect the polymerization reaction, such as heat transfer, monomer diffusion and absorption, and particle size growth. However, the system is always dominated by the polymerization mechanism and supplemented by other microsphere polymerization reactions. Although the proportion of side reactions can be reduced by optimizing the polymerization conditions, in practice, the side reactions of microspheres cannot be avoided. For example, in dispersion polymerization, the system is initially homogeneous, but the primary nuclei form quickly in the system. Then, the microspheres form with the primary nuclei as the center, and then the particle size increases step by step. There are always secondary particles of small particle size in the synthesized microspheres; moreover, since the monomer is in excess, the system also tends to form monomer droplets and polymer particles with a very large particle size and wide distribution in a similar way to suspension polymerization. Therefore, the synthesized microspheres always require several separations to obtain polymer microspheres of uniform particle size. The monomer conversion rate is generally 70–90%, so more than 10% of the monomers generate microspheres of large or small particle size by other mechanisms; after the reaction, some of the large sized microspheres and agglomerates of microspheres can be easily separated and removed by sieving, and more of the very small sized microspheres can be removed in the process of centrifugal washing, but microspheres of 500 nm–50 µm are difficult to separate and purify. The preparation process of this study is shown in Figure 1A, where we first prepared the microsphere crude products using dispersion polymerization (Figure 1B), and then further screened the PS microspheres using the gradient sedimentation method (Figure 1C) and a 650 nm ultrafiltration centrifuge tube (Figure 1D). As shown in Figure 1C,D, the particle size distributions of the PS microspheres screened by gradient sedimentation and 650 nm ultrafiltration centrifuge tubes gradually narrowed.

### 3.2. Fluorescence Spectra and Particle Size Distribution of f-PS Microspheres and SA

#### 3.2.1. Comparison of Fluorescence Spectra of SA and f-PS Microspheres

The fluorescence spectra of f-PS microspheres and SA showed that f-PS microspheres produced a wide fluorescence emission peak in the range of 400–500 nm, with the maximum fluorescence emission peak stabilizing at 439 nm and 462 nm, respectively. Meanwhile, SA produced a wide fluorescence emission peak in the range of 400–600 nm, with the maximum fluorescence emission peak stabilizing at 465 nm (Figure 2A). We found that SA and f-PS microspheres possessed similar maximum fluorescence emission peaks and fluorescence emission peak ranges. We further characterized the change in the fluorescence intensity of f-PS over six months; the results showed that f-PS fluoresced most strongly at day 0 (0 d), and the fluorescence intensity stabilized after 60 d, demonstrating the stability of the fluorescent microspheres (Figure 2B).

#### 3.2.2. Comparison of Particle Size Distribution of SA and f-PS Microspheres

We compared the particle size distributions of f-PS microspheres and SA (Figure 3A). We found that different sonication times (10–40 min) had significant differences on the particle size distribution of SA (Figure 3B–H), and the particle size of SA became smaller as the sonication time gradually increased. We further observed the morphological characteristics of SA by scanning electron microscopy for the corresponding sonication times. It was found that although the particle size of SA was still becoming smaller after 35 min, the photographs of SEM showed that the morphology of SA was heavily disrupted and the amount of SA was drastically reduced with the increase in the sonication time. Therefore, for the subsequent experiments, we adopted a sonication time (30 min) with better monodispersity and morphological characteristics of SA. Although the adjustment of the sonication time gave a better perception of the particle size of the AuP, there was still a gap between the images and those we observed under the microscope and SEM. The characterization results of the nanosizer showed that although the particle size distributions of the f-PS microspheres and SA tended to be similar after becoming sufficiently dispersed, there was still a slight difference. We further found that some of the SA was connected together (the red circle part in the Figure 3), using electron microscope picture characterization. Due to the detection principle of the nanoparticle sizer, the red-circled portion in Figure 3 is considered as a unit statistic, resulting in a biased detection of the average particle size of SA. Therefore, we further investigated the SA aerosol.

### 3.3. Comparison of Particle Size Distribution of SA and f-PS Microspheres in Aerosols

The sampler used in this study was designed based on the Anderson impact principle, which can achieve the classification screening of particles of different particle sizes through the impact effect of particles and the aerodynamic principle. The experimental design idea of this study was to atomize monodisperse aerosol particles of different particle sizes to form an aerosol environment with uniform and stable concentrations. The sampler was placed in a measurement chamber and the stability of the aerosols in the mixing chamber was monitored with an aerodynamic particle size spectrometer until the mean deviation of the bioaerosol concentration in the chamber within 2 min did not exceed ±5% of the set concentration, and the experiment could be conducted.

Dust generation was carried out with f-PS microspheres of 800 nm particle size (Figure 4A–C). During the experiment, the flow rate of the aerodynamic particle size spectrometer was controlled by shunting to keep it consistent with the instrument under test (Figure 4D,E). To verify the accuracy of the calibration results of the f-PS microspheres, SA with a particle size close to it was selected, incubated overnight, centrifuged to collect bacteria, washed 5–10 times, mixed with ultrapure water, sonicated for 30 min, and fumed by nebulization to produce bioaerosols. The control groups for this experiment were air and ultrapure water, and it was found that the particle sizes of the control groups were mainly concentrated in the range of 100 nm–300 nm, whereas the particle sizes of most of the f-PS microspheres and SA were concentrated in the range of 500 nm–1000 nm (Figure 4A–C). The aerosol results show that SA exists in air in a form that is similar to the f-PS properties we have provided.

### 3.4. Fluorescence Lifetimes of f-PS Microspheres

To verify the stability of the f-PS microspheres, the fluorescence lifetime of the f-PS microspheres was examined at 30 d intervals using steady-state/transient fluorescence spectrometry. The results showed that the fluorescence lifetimes of the f-PS microspheres were slightly reduced, but not significantly different from those of f-PS microspheres over 0–180 d, which were all in the range of 3.5 ns–4 ns (Figure 5). From the experimental results, it can be seen that the f-PS microspheres prepared in this study have good stability when used for bioaerosol monitor calibration; the experimental results were easily reproducible, and the preparation process was simple and easy to preserve.

## 4. Discussion

Early market research by the group showed that PS microspheres produced by Thermo Fisher could have particle size errors as accurate as 1% and a solid content of 1%. In contrast, most of the microspheres sold by dozens of Chinese biotechnology companies mainly engaged in microsphere production have a particle size variation factor of 5–15% and a solid content of 5–10%. The PS microspheres sold on the market have a single size specification; most of the companies do not sell PS microspheres with the above size specification, some of them have a high coefficient of variation (>10%), and the remaining few companies need to customize them, which is not only a requirement for the order quantity, but also costly and requires at least 1–2 months of waiting time.

SA, one of the most notorious and prevalent bacterial pathogens, is a major causative agent of pneumonia and other respiratory tract infections [36]. Due to its high frequency of infection, it has gradually become a considerable public health burden [37]. The real-time detection of airborne bioparticles has been achieved [33,34]. However, when calibrating SA aerosol monitoring instruments, it is necessary not only to identify and count SA, but also to calibrate the particle size. This requires that the standards prepared must not only be able to be excited to fluorescence at a specific wavelength, but must also be monodisperse in terms of particle size. To date, there have been few studies of f-PS microspheres with single simulated SA fluorescence properties. In this study, monodisperse polystyrene microspheres with an average particle size of about 0.8 μm were prepared by dispersion polymerization and further screened for the prepared PS microspheres, followed by identifying better monodisperse SA by ultrasonic time discrimination. From comparing the particle size distribution and fluorescence spectra, we found that the fluorescence characteristic peak of SA was at 462 nm, while the monodisperse f-PS microspheres had a fluorescence characteristic peak at 465 nm; the particle size distribution of SA was 700 nm–1000 nm, while the particle size distribution of f-PS microspheres was 780 nm–1000 nm. We also examined the fluorescence lifetime of the f-PS microspheres and found that they still possessed similar fluorescence properties after 180 ds. We further verified that the prepared monodisperse f-PS microspheres had similar particle size distribution with SA in aerosol, mainly in the range of 500 nm–1000 nm; meanwhile, we found that the existing SA detection technique still has the possibility of false detection if the sonication time is not defined. In this study, we refined the fluorescence characteristics and particle size distribution of SA based on the definition of ultrasound time, so that SA could be accurately identified.

## 5. Conclusions

In this study, monodisperse f-PS microspheres with diameters and fluorescence properties similar to those of SA were obtained using the dispersion polymerization method and series screening technique, and a SA simulator based on f-PS microspheres and its preparation method were proposed. The simulants described in this study can effectively simulate SA in terms of particle size, fluorescence characteristic signal, and aerosol simulation, which provides a new idea in the simulation testing of bioaerosol monitoring and protective equipment. In addition, we have improved the fluorescence characteristics and particle size distribution of SA according to the sonicated time, so that SA can be accurately recognized.

## Figures and Tables

**Figure 1 polymers-15-03614-f001:**
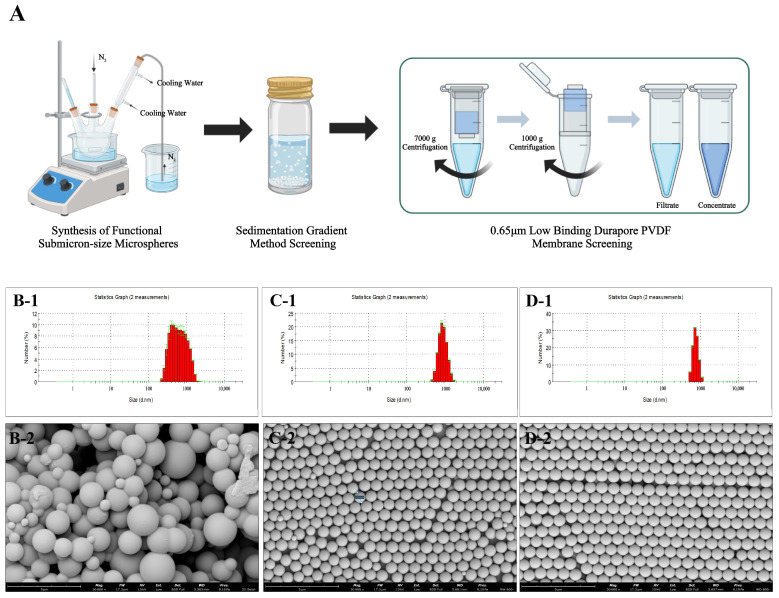
Effect of different sieving methods on particle size dispersion of microspheres. (**A**): Schematic diagram of microsphere screening; (**B**) (**B-1**,**B-2**): PS microsphere crude product; (**C**) (**C-1**,**C-2**): gradient sedimentation method for sieving microspheres; (**D**) (**D-1**,**D-2**): gradient sedimentation method and 650 nm ultrafiltration centrifuge tube for sieving microspheres.

**Figure 2 polymers-15-03614-f002:**
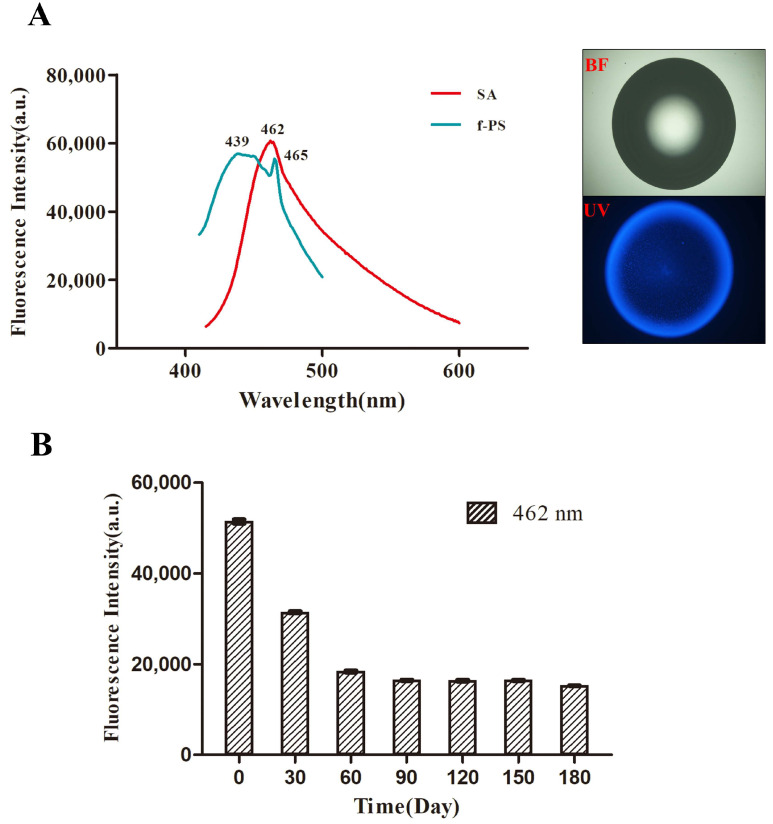
Fluorescence emission spectra of SA and f-PS microspheres (**A**) and fluorescence intensity of f-PS from 0 d–180 d (**B**) BF: bright field.

**Figure 3 polymers-15-03614-f003:**
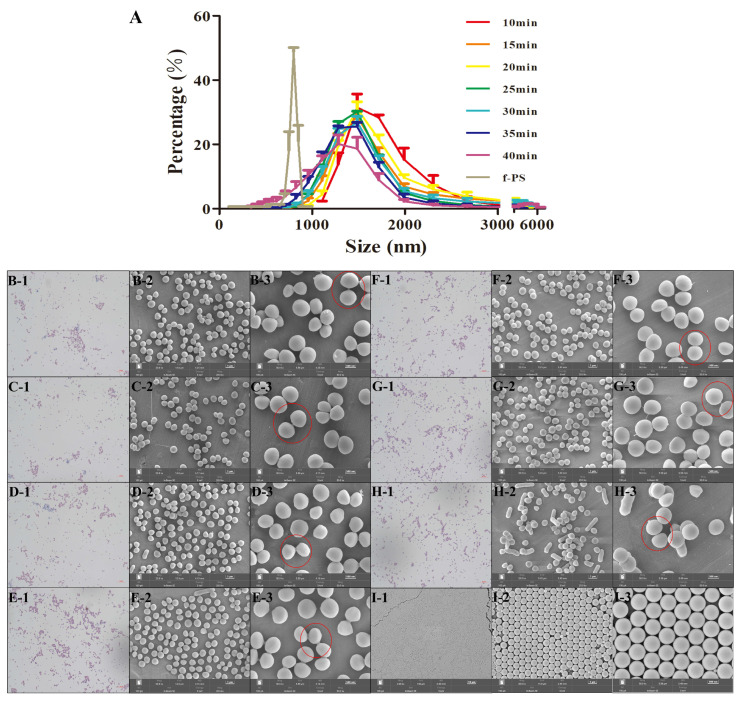
Effect of different sonication times on the particle size distribution of SA. Particle size distribution (**A**) and SEM images of SA (**B**–**H**) and f-PS microspheres (**I**). (**B**–**H**): SEM images of SA at different sonication times (**B**: 10 min, **C**: 15 min, **D**: 20 min, **E**: 25 min, **F**: 30 min, **G**: 35 min, **H**: 40 min). (**B–I**)**-1**: 1000× optical microscope image, (**B–I**)**-2**: 10,000× SEM image, (**B–I**)**-3**: 20,000× SEM image. Red circle: Some of SA are connected together.

**Figure 4 polymers-15-03614-f004:**
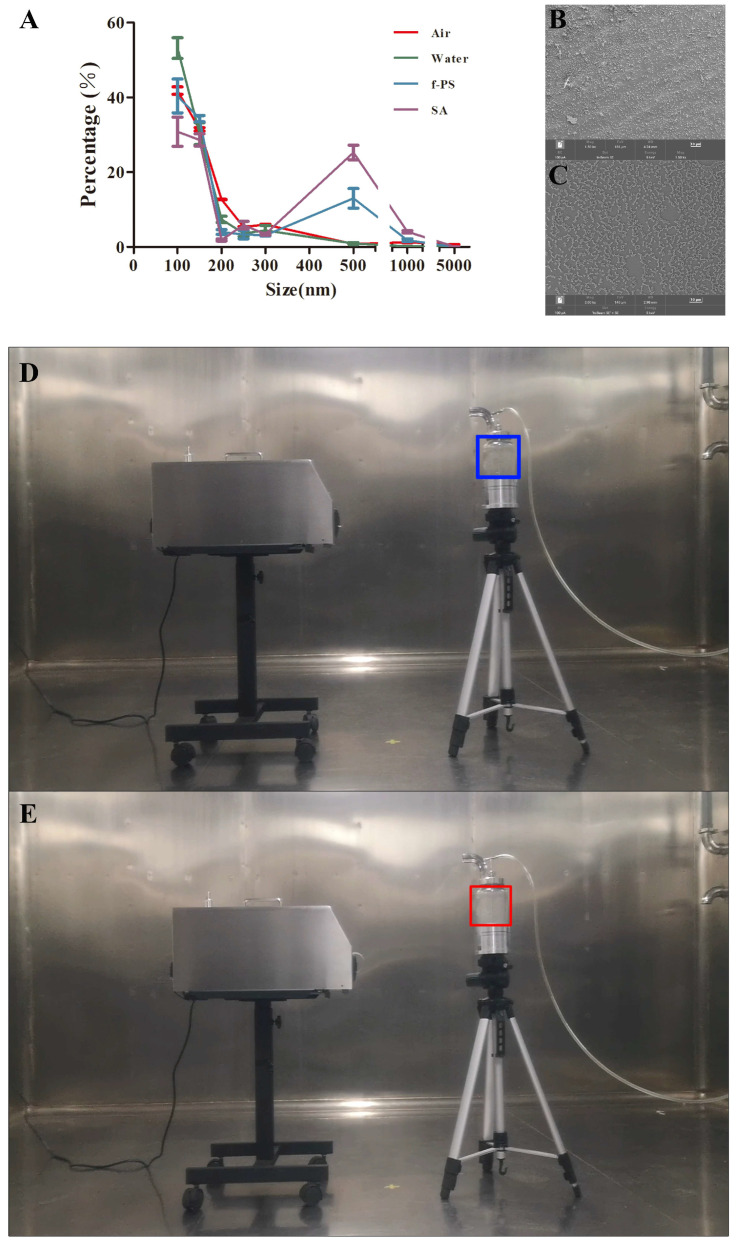
Particle size distribution (**A**) and SEM images of SA (**B**) and f-PS microspheres (**C**); aerosol initiation (blue box in **D**) and onset (red box in **E**).

**Figure 5 polymers-15-03614-f005:**
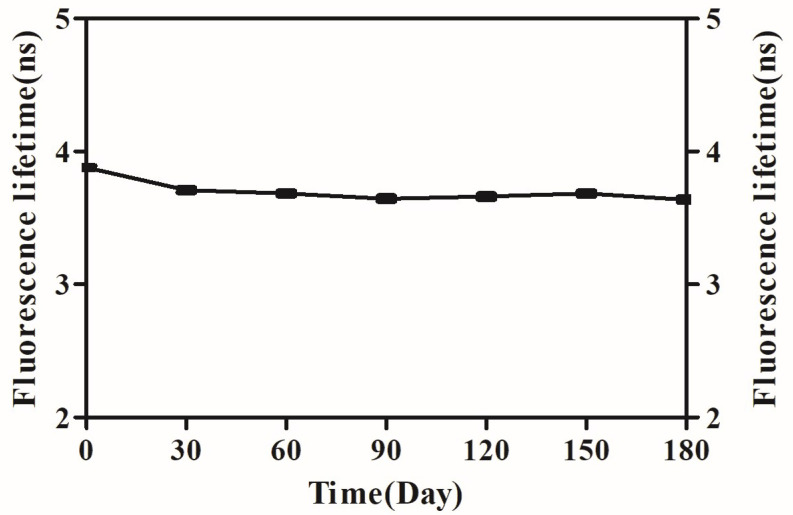
Fluorescence lifetimes of f-PS microspheres from 0 d–180 d, N = 3.

**Table 1 polymers-15-03614-t001:** Effect of different St concentrations on the particle size of microspheres.

	St			PVP			AIBN			H_2_O			Temperature		
	6%	10%	14%	1%	3%	5%	0.6%	1%	1.4%	0	10%	20%	60 °C	70 °C	80 °C
Size (nm)	560	722	779	1015	722	659	516	722	964	329	722	821	904	722	887
CV (%)	13.6	15.8	18.7	19.8	15.8	12.9	18.7	15.8	14.9	18.5	15.8	16.8	18.9	15.8	17.9

## Data Availability

The data that support the findings of this study are available from the corresponding author Tian F upon reasonable request.

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
