# Peer review of "Monodisperse Fluorescent Polystyrene Microspheres for Staphylococcus aureus Aerosol Simulation"

_polymers, 2023, doi:10.3390/polym15173614_

Round 1

Reviewer 1 Report

The manuscript describes the synthesis of fluorescent polystyrene microspheres capable of simulating the properties of Staphylococcus Aureus aerosols. It is interesting and merits publication after addressing the following issues. The language of the manuscript is acceptable. There are still some errors though that need rechecking

11)      Bacteria names should be in italics.

22)      Page 4 lines 113-114: “the volume ratio of water to ethanol in said ethanol solution was m:36” What do the authors mean by m:36?

33)      The authors claim three different reaction temperatures n °C (n=60,70,80) and three different volume ratios of water to ethanol (m=0,4,9). I suppose 0 means pure ethanol. Specifications of this solvent are needed. I do not imagine that the authors performed reactions beyond the solvent's boiling point (78.37 °C).

44)        I do not see any good reason for Figures S1-S5 not to be included in the main manuscript. They are beautiful and will increase its quality.

55)      In Table 1 there are 5 parameters St PVP and AIBN concentrations H2O and Temperature and we get only 15 microsphere sizes (instead of 35 = 243). This most probably means that when one variable changes the others remain stable. What are the values of these variables? Furthermore, the legend “Effect of different St concentrations on the particle size of microspheres” is inaccurate and adds to the confusion.

66)      The legend in Figure 3 is also inadequate.

77)      The Conclusions section needs to be extended.

The language of the manuscript is acceptable. There are still some errors though that need rechecking

Author Response

Dear reviewer

On behalf of all the contributing authors, I would like to express our sincere appreciations of your letter and reviewer’s constructive comments concerning our article entitled “Monodisperse fluorescent polystyrene microspheres for Staphylococcus aureus aerosol simulation” (Manuscript ID: polymers-2556888). These comments are all valuable and helpful for improving our article. According to the editor and reviewer’s comments, we have made extensive modifications to our manuscript. In this revised version, changes to our manuscript were all highlighted within the document by using yellow-colored text. Point-by-point responses to the nice editor and nice reviewer are listed below this letter.

Kind regards,

Prof. Tian

Reviewer 2 Report

Reviewers' comments:

Manuscript Number: polymers-2556888

Full Title: Monodisperse fluorescent polystyrene microspheres for Staphylococcus aureus aerosol simulation.

Comments: 

The manuscript reported on Monodisperse fluorescent polystyrene microspheres for Staphylococcus aureus aerosol simulation. The manuscript needs a detailed editing. It cannot be recommended for publication in the present form. I hope the following points would be helpful for the authors.

- Qualitative information’s are missing in abstract.

- Add more suitable keywords.

- The introduction is very poor and less informative. Authors should elaborate their introduction section by citing few more relevant references. The novelty of the work should also be highlighted.

- The Materials and Methods section should be detailed especially for the 2.2.3. Characterization of Functional Submicron-Size Microspheres, and 2.2.4. Fluorescence Spectrum and Particle Size Distribution of SA. 

- 3.2.2. Comparison of Particle Size Distribution of SA and f-PS Microspheres – should be improve.

- 3.4. Fluorescence Lifetimes of f-PS Microspheres – should be improve.

- Conclusions: authors need to improve.

- References: there are recent references in 2022-2023 treating the same subject, you can use.

- Make all references in same format for volume number, page numbers and journal name, because it is difficult to searching and reading.

- Some sentences need reconstruction and the level of English should be improved.

Based on these, I advise the authors to rectify the above-mentioned errors and we hope to re-evaluate the revised manuscript.

Some sentences need reconstruction and the level of English should be improved.

Author Response

(The authors gave the same response as above.)

Round 2

Reviewer 2 Report

 Now the paper is suitable for publication in this journal. It can be accepted.

Minor editing of English language required.